# Electron-Impact Excitation of the $\lambda$190.8 nm and $\lambda$179.9 nm Intercombination Lines in the Tl$^+$ Ion

Anna Gomonai [1,*], Viktoria Roman [1] , Aleksandr Gomonai [1], Aloka Kumar Sahoo [2] and Lalita Sharma [2]

1 Institute of Electron Physics, Ukrainian National Academy of Sciences, 21 Universitetska Str., 88017 Uzhhorod, Ukraine
2 Indian Institute of Technology Roorkee, Roorkee 247667, India
* Correspondence: annagomonai@gmail.com

**Abstract:** The results of experimental and theoretical studies on electron-impact excitation of the $6s6p\ ^3P_1^\circ \to 6s^2\ ^1S_0$ ($\lambda$190.8 nm) and $6s7s\ ^1S_0 \to 6s6p\ ^3P_1^\circ$ ($\lambda$179.9 nm) intercombination transitions in the single-charged thallium ion are presented. The crossed-beams technique was used in combination with a spectroscopic method in the experiment. A distinct structure revealed in the cross-sections of both lines results from electron decay of atomic autoionizing states and radiative transitions from upper ionic levels. The dominant mechanism of the structure formation was the Coster–Kronig process. Relativistic distorted wave calculations were performed to obtain emission cross-sections for the above transitions. The absolute values of the cross-sections were found to be $(0.25 \pm 0.08) \times 10^{-16}$ cm$^2$ ($\lambda$190.8 nm) and $(0.10 \pm 0.04) \times 10^{-16}$ cm$^2$ ($\lambda$179.9 nm) at the electron energy of 100 eV.

**Keywords:** cross-sections; electron-impact excitation; distorted wave functions; crossed beams; multi-configuration Dirac–Fock method

## 1. Introduction

Elementary processes involving collisions of electrons with positive ions are of significant interest to astrophysicists, who use atomic data, particularly excitation cross-sections, to model conditions in non-equilibrium plasma of stellar atmospheres. Most of the emission lines observed in astrophysical studies result from the excitation of positive ions by electrons. Research in recent years has revealed lines of heavy elements in addition to those of light elements in the spectra of the Sun and stars. In particular, the $\lambda$132.2 nm ($6s6p\ ^1P_1^\circ \to 6s^2\ ^1S_0$) resonance and $\lambda$190.8 nm ($6s6p\ ^3P_1^\circ \to 6s^2\ ^1S_0$) intercombination lines of singly ionized thallium were detected in the spectra of a chemically peculiar HgMn star $\chi$ Lupi [1–4]. Therefore, studying electron-impact excitation cross-sections of these spectral lines is of practical interest. In addition, the Tl$^+$ ion, being a multi-electron ionized system with two valence electrons, is also interesting for atomic physics owing to the great interest in the study of multi-electron interactions and their appearance in various elementary processes observed in recent decades. From a theoretical point of view, the Tl$^+$ ion is a good candidate for testing different models to obtain a correct account of relativistic and correlation effects, which cannot be considered independently of one another in such a heavy ion [5], in order to describe the interaction of heavy metal ions with slow electrons.

The probability of intercombination lines resulting from spin-exchange transitions is significant for multi-electron ions due to the strong relativistic and correlation effects. It is worth noting that the contribution of resonance processes, related to the formation and Auger decay of autoionizing states (AIS), to the emission cross-section dominates over direct excitation [6]. Information on the role of AIS in the dynamics of electron–ion collision process, on the one hand, is a source of data on the structure of complex atomic systems, which allows one to select theoretical models more carefully while, on the other hand, having significant practical applications.

So far, there have been few works in the literature devoted to the study of the electron-impact-induced excitation of the intercombination transitions for single-charged metal ions. The first experiment was related to the measurement of the absolute emission cross-section for the process e + Li$^+$ $\left(1s^2\,^1S_0\right)$ → e+Li$^+$ $\left(1s2p\,^3P_1^\circ\right)$ → $h\nu$ ($\lambda$548.5 nm) [7]. The results obtained in that work gave impetus to the theoretical studies on resonance effects in the excitation cross-section of the Li$^+$ ion [8]. In Ref. [9], calculations of collision strengths for intercombination transitions in the Al$^+$ ion were performed using the R-matrix method. These results show that the resonance processes due to electronic decay of atomic AIS, especially in the near-threshold region of incident electron energies, make the main contribution to the excitation of the intercombination transitions in the Al$^+$ ion. Our first measurements of the relative cross-section of the $6s6p\,^3P_1^\circ \to 6s^2\,^1S_0$ intercombination transition in the Tl$^+$ ion also revealed a complex structure [10].

This work is a continuation of our previous study on the absolute cross-sections of the resonance $6s6p\,^1P_1^\circ \to 6s^2\,^1S_0$ ($\lambda$132.2 nm) and cascade $6s7s\,^1S_0 \to 6s6p\,^1P_1^\circ$ ($\lambda$309.2 nm), $6p^2\,^1D_2 \to 6s6p\,^1P_1^\circ$ ($\lambda$150.8 nm) spectral transitions in the Tl$^+$ ion [11]. Here, we present the results of experimental and theoretical studies on electron-impact excitation of the $6s6p\,^3P_1^\circ \to 6s^2\,^1S_0$ ($\lambda$190.8 nm) and $6s7s\,^1S_0 \to 6s6p\,^3P_1^\circ$ ($\lambda$179.9 nm) intercombination transitions:

$$
\begin{aligned}
e + \text{Tl}^+ \left(6s^2\,^1S_0\right) &\to \text{Tl}^{+*}(6s6p\,^3P_1^\circ) + e' \\
&\downarrow \\
&\text{Tl}^+ \left(6s^2\,^1S_0\right) + h\nu_1\ (\lambda 190.8\ \text{nm})
\end{aligned}
\tag{1}
$$

$$
\begin{aligned}
e + \text{Tl}^+ \left(6s^2\,^1S_0\right) &\to \text{Tl}^{+*}(6s7s\,^1S_0) + e' \\
&\downarrow \\
&\text{Tl}^+ \left(6s6p\,^3P_1^\circ\right) + h\nu_2\ (\lambda 179.9\ \text{nm})
\end{aligned}
\tag{2}
$$

Note that no data on Process (2) are available in the literature so far.

## 2. Experiment

A simplified schematic representation of the experimental setup used in this work is presented in Figure 1. The main components of the experimental apparatus, as well as the measurement procedure, are described in detail elsewhere [12].

Beams of Tl$^+$ ions and monoenergetic electrons are crossed at the right angle in an equipotential collisional region at the ambient pressure of $7 \times 10^{-6}$ Pa. The emission resulting from the radiative transitions under study is detected orthogonally to the plane of the crossing beams.

A low-voltage ($U < 10$ V) arc discharge source is used to obtain a collimated beam (with a cross-section area of $2.5 \times 2.5$ mm$^2$) of single-charged Tl$^+$ ions mainly in the ground state with the current of $0.8 \times 10^{-6}$ A at the ion energy of 1 keV. A metallic thallium sample with 99.9% purity is used. The discharge voltage $U$ is chosen so as to prevent the formation of Tl$^+$ ions in the long-lived $6s6p\,^3P_{0,2}^\circ$ states which can reach the collision region and significantly contribute to the background when measuring the excitation from the ground $6s^2\,^1S_0$ level. A low-energy three-electrode electron gun is used to form a ribbon ($1 \times 8$ mm$^2$) monoenergetic (FWHM $\approx$ 0.5–1.0 eV) electron beam with the current of $(10–150) \times 10^{-6}$ A in the energy range of 5–300 eV. The usage of the ribbon electron beam allows us to increase the emission signal intensity due to the enlargement of the collision region. The electron beam is fully positioned within the ion beam cross-section. The electrons that passed through the collision region are detected by coaxially positioned external and internal Faraday cups made in the form of parallelepipeds with entrance apertures of $3 \times 12$ mm$^2$ and $2 \times 10$ mm$^2$, respectively. This allows us to monitor the stable alignment of the electron beam within the ion beam with the electron energy change by the constancy of the ratio of the currents to each of the cups. A noncommercial VUV monochromator based on the Seya–Namioka scheme with a reciprocal linear dispersion of

1.7 nm/mm and a commercial solar-blind FEU-142 photomultiplier are used to analyze and detect the emission from the collision region.

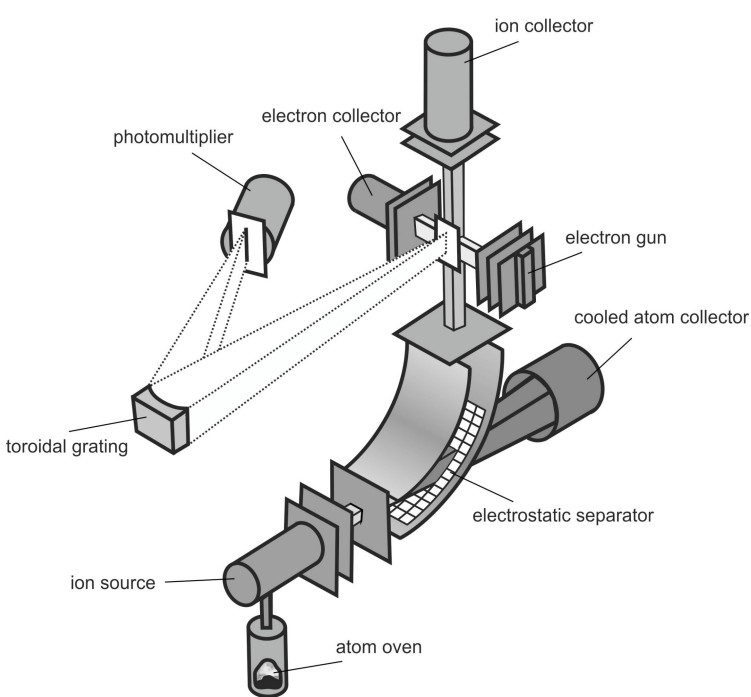

**Figure 1.** Schematic representation of the experimental setup.

It is worth noting that the emission signals originating from the processes under investigation are observed against a large background resulting from the interaction of both beams with the residual gas. For this reason, we use modulation of the electron and ion beams with rectangular pulses shifted with respect to each other by a quarter of a period in combination with a four-way chopping registration system. This allows us to obtain the useful signal intensity of 1–2 counts per second at a signal-to-background ratio of 0.05–0.1 for the $\lambda$190.8 nm line at an electron monoenergeticity of 0.5 eV. As for the $\lambda$179.9 nm line, the useful signal intensity does not exceed 0.5 counts per second at the signal-to-background ratio of 0.03–0.05. For this reason, measurements are performed at an inferior electron monoenergeticity of 1 eV in order to increase the intensity of the useful signal up to 1–2 counts per second. The electron monoenergeticity depends on the accelerating voltage at which the electron gun is adjusted. It is 7 V (FWHM $\approx$ 0.5 eV) and 10 V (FWHM $\approx$ 1 eV). The adjustment is to set appropriate voltages on the first and second anodes of the electron gun so that to obtain the maximum electron current simultaneously with a current–voltage characteristic rapidly increasing and saturating as soon as possible. A retarding potential technique is used to determine the electron monoenergeticity.

The electron beam energy and monoenergeticity are determined with uncertainty of not more than 0.1 eV. The electron energy scale is calibrated according to the excitation threshold of the $\lambda$121.6 nm line of atomic hydrogen by electron impact. The calibration uncertainty of the spectral sensitivity of the detection system, determined on the basis of the emission intensities of the atomic nitrogen spectral lines [13] resulting from the electron impact of $N_2$ molecules measured at the electron energy of 100 eV, is about 16%.

The experimental uncertainties are dominated by the statistical uncertainties of counting. Therefore, the uncertainty of the useful signal measurement is evaluated with a standard uncertainty using the method of evaluation of uncertainty by the statistical analysis of a series of observations. All statistical uncertainties are quoted at the 68% confidence level (CL), corresponding to the mean standard deviation. The uncertainty of the relative measurement does not exceed 15% and 25% for the $\lambda$190.8 nm and $\lambda$179.9 nm lines,

respectively. The total standard uncertainty of the absolute emission cross-section determination, involving the uncertainties of the relative measurement and the spectral sensitivity calibration, does not exceed 31% ($\lambda$190.8 nm) and 41% ($\lambda$179.9 nm).

## 3. Theoretical Method

We used the relativistic distorted wave (RDW) approximation, a perturbative approach, to study the electron-impact excitation of singly ionized thallium ion. The emission cross-sections are calculated for the $\lambda$190.8 nm ($6s6p\,^3P_1^\circ \rightarrow 6s^2\,^1S_0$) and $\lambda$179.9 nm ($6s7s\,^1S_0 \rightarrow 6s6p\,^3P_1^\circ$) intercombination lines. The present RDW approach is similar to that employed in our previous work [11] on obtaining electron impact cross-sections of the $6s6p\,^1P_1^\circ \rightarrow 6s^2\,^1S_0$, $6s7s\,^1S_0 \rightarrow 6s6p\,^1P_1^\circ$, and $6p^2\,^1D_2 \rightarrow 6s6p\,^1P_1^\circ$ transitions in the Tl$^+$ ion.

For the cross-section calculations, we obtained atomic bound state wavefunctions in the multiconfiguration Dirac–Fock approach using the GRASP2018 package [14]. In the MCDF method, the atomic wavefunctions $\Psi$ are considered as a basis of configuration state functions (CSFs) having the same parity $P$ and total angular quantum number $J$:

$$\Psi(\gamma PJM) = \sum_{i=1}^{n} c_i \phi_i(\gamma_i PJM) \tag{3}$$

where $\phi_i$ represents the CSFs, which are an antisymmetrized product of the orthonormal set of Dirac orbitals, and $c_i$ represents the mixing coefficients. $\gamma$ denotes all the quantum numbers required for unique representations of the CSFs. In the present study, we followed a restricted active space approach with single and double excitations from the multireference set {$6s^2$, $6s6p$, $6s7s$} to the desired active space which is expanded up to {$12s$, $11p$, $10d$, $7f$, $6g$}. We have also considered a maximum of two electron excitation from the $5d^{10}$ subshell to include the crucial core-valence correlations in the atomic structure calculations. The radial wavefunctions and the expansion coefficients are obtained by optimizing them with the relativistic self-consistent method. Relativistic configuration interaction calculations are performed following the MCDF calculations, which also include the Breit interaction and quantum electrodynamic corrections.

To establish the accuracy of the present MCDF atomic wavefunctions, we compared our calculated wavelengths and transition rates for the $6s6p\,^3P_1^\circ \rightarrow 6s^2\,^1S_0$ and $6s7s\,^1S_0 \rightarrow 6s6p\,^3P_1^\circ$ transitions with the experimental and theoretical data available from the literature. Our calculated wavelengths for the above two transitions are 189.6 nm and 177.04 nm, respectively, which are within 1.5% of the measured values [15]. The present transition rate for $\lambda$190.8 nm line is $2.63 \times 10^7$ s$^{-1}$ which shows an excellent match with the measured value ($2.6 \times 10^7$ s$^{-1}$) [16] and other theoretical results ($2.7 \times 10^7$ s$^{-1}$) [17,18]. For the $\lambda$179.9 nm line the transition rate obtained here is $2.51 \times 10^7$ s$^{-1}$. For this transition rate, there are no experimental data available with which to perform a comparison. However, two previous theoretical works have reported its value to be $3.3 \times 10^7$ s$^{-1}$ [18] and $8.7 \times 10^7$ s$^{-1}$ [17]. Thus, the three theoretical calculations are of the same order of magnitude. We checked the ratio of the transition rates in the Coulomb and Babushkin gauges and found it to be nearly 1.3. Thus, a good agreement between the results from the two gauges indicates the quality of the wavefunctions.

Having confirmed that the calculated atomic wavefunctions are appropriate, we solved the Dirac equations using the spherically symmetric static potential of the Tl$^+$ ion and obtained the continuum wavefunctions of the projectile electron [19]. Finally, the RDW T-matrix is evaluated using the bound and continuum states wavefunctions to calculate the electron impact cross-section for the excitation from an initial state $a$ to a final state $b$ [19], i.e.,

$$\sigma_{a \rightarrow b} = (2\pi)^4 \frac{k_b}{2(2J_a + 1)k_a} \sum_{M_a \mu_a M_b \mu_b} \int \left| T_{a \rightarrow b}^{RDW}(\gamma_b, J_b, M_b, \mu_b; \gamma_a, J_a, M_a, \mu_a) \right|^2 d\Omega \tag{4}$$

where $k_{a(b)}$ is the projectile electron wave vector, $J_{a(b)}$ and $M_{a(b)}$ represent the total angular momentum quantum number and the corresponding magnetic quantum number, $\mu_{a(b)}$ denotes the spin projection of the projectile electron. The subscripts $a$ and $b$ represent the initial and final channels, respectively.

In the atomic wavefunction calculations, the GRASP2018 gives the radial wave functions up to 1990 points using an exponential grid. The radial wavefunctions are further interpolated at 7000 mesh points using the RDW program and thereafter, distorted wavefunctions for projectile and scattered electrons are also calculated at the same 7000 grid points. The RDW program can handle up to 250 partial waves and for a given incident electron energy it uses as many partial waves as are required to achieve the tolerance of $10^{-5}$ in the evaluation of the $T$-matrix. Once convergence is reached, the excitation cross-section is finally determined from Equation (3). More details of the RDW method can be found in [19].

Furthermore, to calculate the cross-sections, first, we considered the direct electron-impact excitation from the ground state $6s^2\,^1S_0$ to the level of interest, i.e., $6s6p\,^3P_1^\circ$ or $6s7s\,^1S_0$. In the second step, the excitations from the ground state to all the higher levels and their successive radiative decay to $6s6p\,^3P_1^\circ$ or $6s7s\,^1S_0$ are considered. In our case, a total of 100 excited states are considered. The branching fractions are also obtained for the decay from the excited levels having more than one decay channel and these branching fractions are multiplied by their corresponding emission cross-sections from the ground state to that particular excited state. These results are added to the direct excitation cross-sections to get a summed-up cross-section. For the resonance $6s6p\,^3P_1^\circ \to 6s^2\,^1S_0$ ($\lambda$190.8 nm) transition only one radiative decay channel from the upper $6s6p\,^3P_1^\circ$ level is possible. Therefore, this summed-up cross-section gives the final emission cross-section. However, in the case of the $6s7s\,^1S_0 \to 6s6p\,^3P_1^\circ$ ($\lambda$179.9 nm) transition, the upper $6s7s\,^1S_0$ level has two possible radiative decay channels, namely to the $6s6p\,^3P_1^\circ$ and $6s6p\,^1P_1^\circ$ states. Thus, we multiplied the branching fraction of the $\lambda$179.9 nm line, i.e., 0.11, by its corresponding summed-up cross-section to obtain the final emission cross-section.

## 4. Results and Discussion

### 4.1. Experimental Excitation Functions

Experimentally measured energy dependences (excitation functions) of the electron impact emission cross-sections of the $\lambda$190.8 nm ($6s6p\,^3P_1^\circ \to 6s^2\,^1S_0$) and $\lambda$179.9 nm ($6s7s\,^1S_0 \to 6s6p\,^3P_1^\circ$) intercombination lines for the Tl$^+$ ion are presented in Figure 2 by the 190.8 nm (exp) and 179.9 nm (exp) curves. A pronounced structure is observed in both functions. However, in the case of the $\lambda$190.8 nm line, it is much more complicated.

Before proceeding to the analysis of the observed structure, we refer to the features of the Tl$^+$ ion energy spectrum. (i) Its ground-state configuration is $5d^{10}6s^2\,^1S_0$. Excitation of an $s$ electron results in the formation of singlet and triplet terms of the $5d^{10}6snl$ ($l = s, p, d, f, g$) configuration [20,21]. (ii) Simultaneous excitation of both $s$ electrons, which is rather effective, gives rise to the formation of the $5d^{10}6p^2$ states. The known $^3P_{0,1,2}$, $^1D_2$ [20], and $^1S_0$ [22] terms lie below the Tl$^+$ ionization potential. According to the selection rules, their decay to the ionic $6s6p$ ($^{1,3}P_J^\circ$) resonance levels is the most probable. (iii) The binding energy of electrons in the subvalence $5d^{10}$ shell is close to that of electrons in the valence $6s^2$ shell [23], resulting in a rather high probability of the $d$ electron excitation leading to the formation of twelve levels of the $5d^9 6s^2 6p$ configuration [20,24,25]. According to Ref. [26], there is a strong configuration interaction of the $5d^9 6s^2 6p$ levels with the $5d^{10}6s7p$ and $5d^{10}6s5f$ levels. As a result, the $5d^9 6s^2 6p$ levels can decay to the ground state and excited even-parity ionic levels. (iv) A great number of atomic AIS converge to each level of the Tl$^+$ ion. All currently known AIS of the Tl atom are located between the first and the second ionization potentials. Data on the energy position of these AIS are available from studies of photoabsorption spectra [24,25,27–29] and ejected-electron spectra [30]. A simplified energy level diagram for the Tl$^+$ ion and Tl atom, including the radiative transitions studied in this work, is shown in Figure 3. The energy values are shown in Table 1.

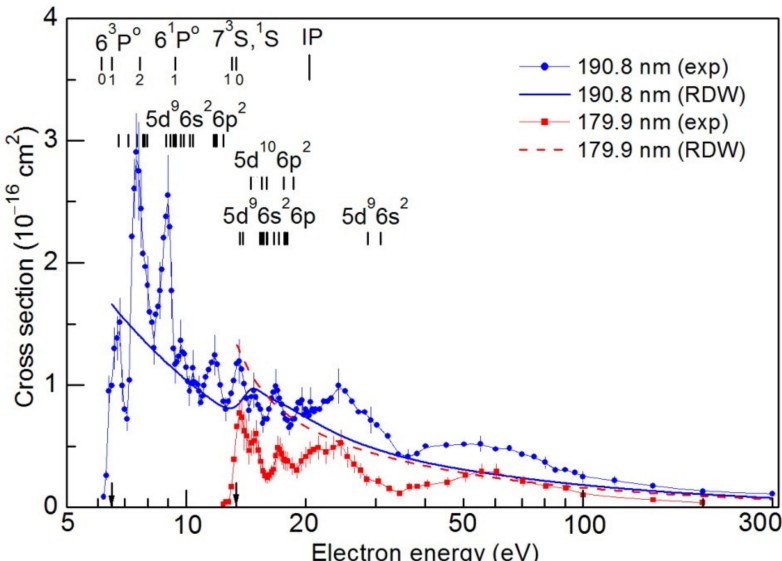

**Figure 2.** Energy dependence of the electron impact emission cross-sections of the $\lambda$190.8 nm ($6s6p\ ^3P_1^\circ \to 6s^2\ ^1S_0$) and $\lambda$179.9 nm ($6s7s\ ^1S_0 \to 6s6p\ ^3P_1^\circ$) intercombination lines of the Tl$^+$ ion. Error bars are the uncertainty of the relative measurements.

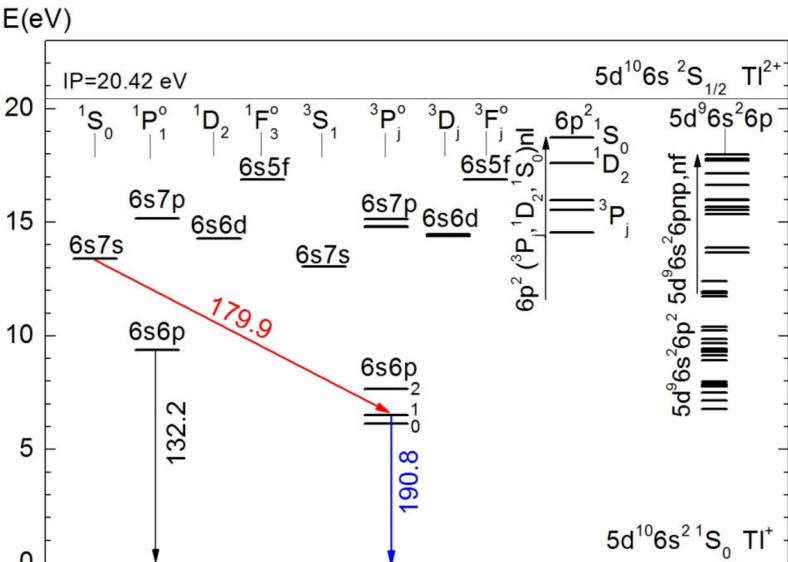

**Figure 3.** Energy level diagram of the Tl$^+$ ion and Tl atom.

Now, we return to the analysis of the observed structure. Each group of maxima in the excitation function of the $\lambda$190.8 nm intercombination line falls within a distinct interval resulting from the presence of ionic states which are the convergence limit of corresponding AIS. In the near-threshold energy range, below the energy of the first excited $6s7s\ ^3S_1$ level ($\approx$13 eV) from which the transition to the $6s6p\ ^3P_1^\circ$ level is already possible, the structure is related to the resonance excitation involving resonance capture of an incident electron with simultaneous excitation of one of the ion electrons (dielectronic capture) resulting in the formation of an atomic AIS, which subsequently decay to an excited state of the ion in the electron channel:

$$e + \mathrm{Tl}^+ \to \mathrm{Tl}^{**} \to \mathrm{Tl}^{+*} + e', \tag{5}$$

where Tl$^{**}$ is an atomic AIS, Tl$^{+*}$ is an excited ionic state.

**Table 1.** Energies of the Tl$^+$ and Tl levels.

| Configuration | Term | J | Energy (eV) | Reference |
|---|---|---|---|---|
| $5d^{10}6s^2$ | $^1S$ | 0 | 0.00 | [20,21] |
| $5d^{10}6s6p$ | $^3P^o$ | 0 | 6.13 | [20,21] |
| | | 1 | 6.50 | [20,21] |
| | | 2 | 7.65 | [20,21] |
| $5d^{10}6s6p$ | $^1P^o$ | 1 | 9.38 | [20,21] |
| $5d^{10}6s7s$ | $^3S$ | 1 | 13.05 | [20,21] |
| $5d^{10}6s7s$ | $^1S$ | 0 | 13.39 | [20,21] |
| $5d^{10}6s6d$ | $^1D$ | 2 | 14.28 | [20,21] |
| $5d^{10}6s6d$ | $^3D$ | 1 | 14.40 | [20,21] |
| | | 2 | 14.43 | [20,21] |
| | | 3 | 14.48 | [20,21] |
| $5d^{10}6s7p$ | $^3P^o$ | 0 | 14.80 | [20,21] |
| | | 1 | 14.82 | [20,21] |
| | | 2 | 15.13 | [20,21] |
| $5d^{10}6s7p$ | $^1P^o$ | 1 | 15.17 | [20,21] |
| $5d^{10}6s5f$ | $^3F^o$ | 3 | 16.87 | [20,21] |
| | | 2 | 16.88 | [20,21] |
| | | 4 | 16.887 | [20,21] |
| $5d^{10}6s5f$ | $^1F^o$ | 3 | 16.89 | [20,21] |
| $5d^{10}6p^2$ | $^3P$ | 0 | 14.56 | [20] |
| | | 1 | 15.54 | [20] |
| | | 2 | 15.97 | [20] |
| $5d^{10}6p^2$ | $^1D$ | 2 | 17.60 | [20] |
| $5d^{10}6p^2$ | $^1S$ | 0 | 18.72 | [22] |
| $5d^96s^26p$ | 1° | 2 | 13.67 | [20,24,25] |
| | 2° | 3 | 13.88 | [24,25] |
| | 3° | 4 | 15.36 | [25] |
| | 4° | 2 | 15.54 | [20,24,25] |
| | 5° | 1 | 15.67 | [20,24,25] |
| | 6° | 3 | 15.96 | [20,24,25] |
| | 7° | 2 | 16.00 | [20,24,25] |
| | 8° | 1 | 16.65 | [20,24,25] |
| | 9° | 0 | 17.14 | [25] |
| | 10° | 3 | 17.70 | [20,24,25] |
| | 11° | 1 | 17.80 | [20,24,25] |
| | 12° | 2 | 17.99 | [20,24,25] |
| $5d^96s^26p^2$ * | | | 6.77 | [30] |
| | | | 7.15 | [30] |
| | | | 7.43 | [30] |

**Table 1.** *Cont.*

| Configuration | Term | $J$ | Energy (eV) | Reference |
|---|---|---|---|---|
| | | | 7.78 | [30] |
| | | | 7.82 | [30] |
| | | | 7.88 | [24,30] |
| | | | 7.99 | [24,30] |
| | | | 8.91 | [30] |
| | | | 9.14 | [30] |
| | | | 9.29 | [30] |
| | | | 9.38 | [30] |
| $5d^9 6s^2 6p^2$ * | | | 9.43 | [25,30] |
| | | | 9.68 | [24,30] |
| | | | 9.86 | [30] |
| | | | 10.23 | [30] |
| | | | 10.39 | [24] |
| | | | 11.73 | [24] |
| | | | 11.86 | [24,25] |
| | | | 11.94 | [25] |
| | | | 12.39 | [25] |
| $Tl^{2+}$ $(5d^{10}6s\ ^2S_{1/2})$ | Limit | | 20.42 | [20] |

* The energies of atomic autoionizing states are shifted by the value of the Tl atom ionization potential (6.11 eV).

Thus, it is known from Refs. [25,28] that the $6s6p\ (^3P_2^\circ,\ ^1P_1^\circ)np\ (n \geq 7)$ AIS are located above the excitation threshold of the $6s6p\ ^3P_1^\circ$ level. These AIS can be an efficient resonance channel of the $6s6p\ ^3P_1^\circ$ level population due to the Coster–Kronig process: $6s6p\ (^{1,3}P^\circ)np \rightarrow 6s6p\ ^3P_1^\circ + e^-$. We recall that the Coster–Kronig transition is a special case of the Auger process in which the vacancy is filled by an electron from a higher subshell of the same shell. As can be seen from Figure 2, this resonance contribution is dominant in the threshold electron energy range appearing in the form of the first three distinct maxima in the energy dependence of the emission cross-section of the $\lambda$190.8 nm intercombination line. According to their energy positions, these maxima fall within the range of the spin-orbit splitting of the $6s6p\ ^3P_1^\circ$ (6.50 eV), $6s6p\ ^3P_2^\circ$ (7.65 eV), and $6s6p\ ^1P_1^\circ$ (9.38 eV) levels [21]. The high probability of AIS decay due to the Coster–Kronig process (from 30% up to 70%) has been confirmed in several studies of AIS formation and decay (see e.g., [31,32]).

The $5d^9(^2D)6s^2 6p^2$ AIS [24,25,30] can also play an essential role due to the Auger decay $5d^9(^2D)6s^2 6p^2 \rightarrow 5d^{10}6s6p(^3P_1^\circ) + e^-$ in the near-threshold resonance excitation of the $\lambda$190.8 nm line in the energy range 6.8–12.4 eV. However, above the excitation energy of the ionic $6s6p\ ^1P_1^\circ$ level (9.38 eV), it is this level to which the decay of such AIS is energetically more probable. This accounts for the low intensity of the maxima at the energies 9.7 eV and 10.4 eV in the excitation function of the $\lambda$190.8 nm line. Unlike these maxima, the maximum at the energy of 11.8 eV is rather distinct. This is due to the fact that not only the $5d^9 6s^2 6p^2$ AIS, but also the $5d^9 6s^2 6p np\ (n \geq 7)$, $5d^9 6s^2 6p nf\ (n \geq 5)$ [24,25,29], $5d^{10}6s7s6d$, and $5d^{10}6s7snl$ [24] AIS contribute to this maximum.

At electron energies above the excitation energy of the $6s7s\ ^3S_1$ ionic level (13.04 eV) [21], the additional channels of the $6s6p\ ^3P_1^\circ$ level population due to the cascade transitions from the $5d^{10}6snl\ ^{1,3}L_J$ ($L = S$, $l = s$, $n \geq 7$; $L = D$, $l = d$, $n \geq 6$), $5d^{10}6p^2\ (^3P_{0,1,2},\ ^1D_2,\ ^1S_0)$, and $5d^9 6s^2 6p$ ionic levels become available.

At energy of 13.39 eV, the ionic level $6s7s\,^1S_0$ [21] can be excited, which may lead to the emission of the $\lambda$179.9 nm line corresponding to the $6s7s\,^1S_0 \to 6s6p\,^3P_1^\circ$ intercombination transition. It can be seen from Figure 2 that although the intensity of the $\lambda$179.9 nm line is lower by a factor of about three compared to that of the $\lambda$190.8 nm line, distinct structures in the excitation function of $\lambda$179.9 nm line can be observed. Moreover, a correlation between the energy positions of the excitation function maxima of both lines suggests the same nature of the structure in both cases. In particular, in the electron energy range from 12.5 eV up to the ion ionization potential (20.42 eV [33]), the observed structure is related mainly to the contribution of the $5d^96s^26pnp$ ($n \geq 7$), $5d^96s^26pnf$ ($n \geq 5$) [24,25,29] and $5d^{10}6p^2np$ ($n \geq 7$), $5d^{10}6p^2nd$ ($n\geq 6$) [25] atomic AIS converging to the $5d^96s^26p$ (13.7–18 eV) [20,25] and $5d^{10}6p^2$ (14.6–18.7 eV) ionic levels [20,22]. As for radiative transitions from these ionic levels, decay of the $5d^96s^26p$ levels to the $6s6p\,^3P_1^\circ$ level is parity forbidden. The allowed transitions are from the $5d^96s^26p$ levels to the $5d^{10}6sns$ and $5d^{10}6snd$ ($n \geq 6$) levels. However, it is worth noting that the $5d^96s^26p$ levels decay to the ground $6s^2\,^1S_0$ ionic state more effectively since this is a single-electron transition when the $6p$ electron fills a vacancy in the $5d$ shell. As for the $5d^96s^26p \to 5d^{10}6sns$, $5d^{10}6snd$ ($n \geq 6$) transitions, these are two-electron transitions when the $6p$ electron fills the vacancy in the $5d$ shell with simultaneous excitation of one of the $6s$ electrons. The probability of such process is much lower than that of the single-electron transition. At the same time, decay of the $5d^{10}6p^2$ levels to the ground $6s^2\,^1S_0$ state is forbidden by parity; however, they effectively decay to the $5d^{10}6s6p$ ionic levels. In particular, the most probable decay channels of the $5d^{10}6p^2\,^3P_{0,1,2}$ levels are the transitions to the $6s6p\,^3P_1^\circ$ state. Since a rather large spin-orbit splitting is characteristic for the $5d^{10}6p^2$ levels, the AIS converging to them decay with a high probability due to the Coster–Kronig effect.

Although the transitions from the $5d^{10}6p^2$ ($^3P_{0,1,2}$, $^1D_2$, $^1S_0$) levels to the $6s7s\,^1S_0$ level are forbidden by the parity selection rules; they can contribute to the population of the $6s7s\,^1S_0$ level via multi-stage cascade transitions. In particular, the $5d^{10}6p^2$ levels effectively decay to the $5d^96s^26p$ levels, which, in turn, can populate the $6s7s\,^1S_0$ state. Simultaneously, the $5d^{10}6p^2np$ and $5d^{10}6p^2nd$ atomic AIS make a direct resonance contribution to the $6s7s\,^1S_0 \to 6s6p\,^3P_1^\circ$ ($\lambda$179.9 nm) transition.

Similar to the resonance $\lambda$132.2 nm line of the Tl$^+$ ion [11], the structures in the excitation functions of the lines under study are also observed above the ionization potential. The structure in the energy range 20.42–42.54 eV is most likely related to decay of the $5d^96s^2np$ ($n \geq 7$), $5d^96s^2nf$ ($n \geq 5$) and $5d^96s6pns$ ($n \geq 6$), $5d^96s6pnp$ ($n \geq 7$) ionic AIS to the $6s6p\,^3P_1^\circ$ and $6s7s\,^1S_0$ level of the Tl$^+$ ion, either directly or via multi-stage cascade transitions. These AIS converge to the $5d^96s^2\,^2D_{5/2}$ (28.67 eV), $5d^96s^2\,^2D_{3/2}$ (30.91 eV) [34], and $5d^96s6p$ (35.46–42.54 eV) levels of the Tl$^{2+}$ ion [34,35]. In the literature, there are only data on the $5d^96s^27p$ AIS located in the energy range 27.12–27.62 eV, while data on the other AIS are not available. However, the fact that such AIS are observed for the mercury atom [36], with respect to which the Tl$^+$ ion is isoelectronic, is an indirect confirmation of our assumption.

Furthermore, the wide maximum in the energy range 50–60 eV results from the processes related to the $5d$ electron detachment with simultaneous excitation of the $6s$ electrons as it occurs, for example, in the case of the In$^+$ ion [37].

Deviation of the excitation functions decreasing at high energies ($\geq$70 eV) from the $E^{-3}$ behavior, typical for intercombination transitions, results from configuration mixing of ionic levels [26].

## 4.2. Absolute Excitation Cross-Sections

The experimentally measured excitation functions of the $\lambda$190.8 nm ($6s6p\,^3P_1^\circ \to 6s^2\,^1S_0$) and $\lambda$179.9 nm ($6s7s\,^1S_0 \to 6s6p\,^3P_1^\circ$) intercombination lines are placed on an absolute scale by comparing them with the absolute cross-section of the $\lambda$132.2 nm ($6s6p\,^1P_1^\circ \to 6s^2\,^1S_0$) resonance line [11], which is $2.28 \times 10^{-16}$ cm$^2$ at an electron energy of 100 eV. For correct comparison, the intensities of all three lines are measured at 100 eV in the same experiment.

Note that the absolute value of the excitation cross-section of the $\lambda 132.2$ nm line in Ref. [11] was obtained by normalizing the experimental data using the RDW cross-section.

It is worth noting that, owing to a rather long lifetime of the $6s6p\ {}^3P_1^\circ$ ($\tau = 39 \pm 2$ ns) and $6s7s\ {}^1S_0$ ($\tau = 3.0 \pm 0.4$ ns) levels [38], a part of excited ions leave the collision region resulting in a decrease in the $\lambda 190.8$ nm and $\lambda 179.9$ nm intensities. Our calculations with using the procedure presented in Ref. [12] show that only 80% ($\lambda 190.8$ nm) and 91% ($\lambda 179.9$ nm) of excited Tl$^+$ ions emit in the collision region. We considered this aspect while determining the absolute values of the excitation cross-section of these lines. Hence, the obtained absolute values are $0.25 \times 10^{-16}$ cm$^2$ ($\lambda 190.8$ nm) and $0.10 \times 10^{-16}$ cm$^2$ ($\lambda 179.9$ nm) at the electron energy of 100 eV.

The results of the RDW calculation are presented in Figure 2 by the 190.8 nm (RDW) and 179.9 nm (RDW) curves. In the case of the $\lambda 190.8$ nm line, the calculated emission cross-section shows an overall good agreement with the experiment. However, the structure is missing in the theoretical cross-section since other inelastic channels, such as autoionization, electron attachment, etc., are not included in the RDW calculations. Furthermore, the calculations are performed in the first order RDW approximation. The value of the cross-section $0.18 \times 10^{-16}$ cm$^2$ at 100 eV is in an excellent agreement with the above-obtained value of $0.25 \times 10^{-16}$ cm$^2$. As for the $\lambda 179.9$ nm line, the theoretical cross-section is consistent with the normalized measurement at high energies ($\geq 60$ eV) while at lower energies the calculation is greater than the experiment. The value of the calculated cross-section $0.16 \times 10^{-16}$ cm$^2$ also matches well with the above-obtained value $0.10 \times 10^{-16}$ cm$^2$ at 100 eV.

It is worth noting that the maximum absolute value of the emission cross-section of the $\lambda 190.8$ nm ($6s6p\ {}^3P_1^\circ \rightarrow 6s^2\ {}^1S_0$) intercombination line is $2.9 \times 10^{-16}$ cm$^2$, which is smaller only by a factor of about three than the corresponding value of $9.7 \times 10^{-16}$ cm$^2$ for the $\lambda 132.2$ nm ($6s6p\ {}^1P_1^\circ \rightarrow 6s^2\ {}^1S_0$) resonance line [11]. This indicates the high probability of the spin-change transitions in the case of heavy ions.

## 5. Conclusions

Electron-impact excitation of the intercombination $6s6p\ {}^3P_1^\circ \rightarrow 6s^2\ {}^1S_0$ ($\lambda 190.8$ nm) and $6s7s\ {}^1S_0 \rightarrow 6s6p\ {}^3P_1^\circ$ ($\lambda 179.9$ nm) spectral transitions from the ground $6s^2\ {}^1S_0$ level in the Tl$^+$ ion is studied. A distinct structure in the near-threshold cross-section of the $\lambda 190.8$ nm line results from the electron decay of the $5d^9 6s^2 6p^2$, $5d^{10} 6s6pnp$ ($n \geq 7$), and $5d^{10} 6s7snd$ ($n \geq 6$) atomic AIS. The dominant mechanism of the structure formation is the Coster–Kronig process related to the decay of the $5d^{10} 6s6pnp$ ($n \geq 7$) AIS converging to the $6s6p\ {}^3P_2^\circ$ and $6s6p\ {}^1P_1^\circ$ ionic levels. A structure above the excitation energy of the $6s7s\ {}^1S_0$ level for both lines is due to the electron decay of the $5d^{10} 6p^2 np$ ($n \geq 7$), $5d^{10} 6p^2 npnd$ ($n \geq 6$), $5d^9 6s^2 6pnp$ ($n \geq 7$), $5d^9 6s^2 6pnf$ ($n \geq 5$) atomic and $5d^9 6s^2 np$ ($n \geq 7$), $5d^9 6s^2 nf$ ($n \geq 5$), $5d^9 6s6pns$ ($n \geq 6$), $5d^9 6s6pnp$ ($n \geq 7$) ionic AIS as well as to the radiative transitions from the higher $5d^{10} 6snl$, $5d^{10} 6p^2$, $5d^9 6s^2 6p$ ionic levels.

The absolute values of the emission cross-section for the transitions studied in this work are determined from the absolute value of emission cross-section for the $\lambda 132.2$ nm resonance line obtained in our earlier study [11] and the experimentally observed intensity ratios for the $\lambda 132.2$ nm, $\lambda 190.8$ nm, $\lambda 179.9$ nm lines. The absolute emission cross-sections are found to be $(0.25 \pm 0.08) \times 10^{-16}$ cm$^2$ ($\lambda 190.8$ nm) and $(0.10 \pm 0.04) \times 10^{-16}$ cm$^2$ ($\lambda 179.9$ nm) at the incident electron energy of 100 eV.

We also carried out the RDW calculations to study the above intercombination transitions. The agreement between the measurements and the RDW calculations for both lines is very good at high energies. However, more inelastic channels should be included to study cascade effects to improve the agreement between the theory and the experiment.

The maximum emission cross-section of the $\lambda 190.8$ nm intercombination line ($2.9 \times 10^{-16}$ cm$^2$) is less than that for the $\lambda 132.2$ nm resonance line ($9.7 \times 10^{-16}$ cm$^2$) [11] only by a factor of about three, which is indicative of the high probability of the spin-change transitions in the case of heavy ions.

**Author Contributions:** Conceptualization, A.G. (Anna Gomonai); software, A.K.S. and L.S.; formal analysis, V.R. and A.G. (Aleksandr Gomonai); investigation, A.G. (Anna Gomonai) and A.G. (Aleksandr Gomonai); writing—original draft preparation, A.G. (Aleksandr Gomonai) and L.S.; writing—review and editing, A.G. (Anna Gomonai), V.R., A.G. (Aleksandr Gomonai), A.K.S. and L.S.; supervision, A.G. (Anna Gomonai); project administration, A.G. (Anna Gomonai); funding acquisition, A.G. (Anna Gomonai). All authors have read and agreed to the published version of the manuscript.

**Funding:** This research received no external funding.

**Data Availability Statement:** Not applicable.

**Conflicts of Interest:** The authors declare no conflict of interest.

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
