# Peer review of "Electron-Impact Excitation of the λ190.8 nm and λ179.9 nm Intercombination Lines in the Tl+ Ion"

_atoms, doi:10.3390/atoms10040136_

Round 1

Reviewer 1 Report

Gomonai et al. report on measurements and calculations of cross sections for electron-impact excitation of two intercombination lines in the Tl+ ion. The topic is of interest in connection with astrophysical observations and their interpretation. Other transitions in Tl+ and other very heavy ions have already been studied by the same group of authors and results have been published. The experiments follow a scheme that has been applied in the Uzhgorod group for many decades. The theoretical calculations are based on the relativistic distorted wave approximation by which direct excitation of the lines under investigation is studied. The measurements show interesting structures in the energy dependences of the cross sections that are associated with a multitude of indirect pathways by which the excited levels can be populated. 

There are some minor issues that should be taken care of and there are some more serious questions that should be addressed before this manuscript is published. In the following I list the points in the sequence in which they occur in the manuscript. I do not point out deficiencies in the presentation in English as long as it is clear what the authors wanted to express.

Line 14: “to” should probably be replaced by “from subsequent(?)”. The sentence is not well phrased and thus, somewhat misleading. When ionic autoionizing states (of Tl+?) undergo electron decay (Auger decay?)  the final charge state is no longer Tl+ and therefore, no contributions from transitions in Tl+ can be observed in that case.

Line 16: somewhere in the text (not necessarily in the Abstract), the term “effective cross section” should be defined

Equation 1: misprint …hhh…

Line 71: rather than describing all aspects of the experiment in detail, the manuscript refers to previous work. This is o.k., in principle. However, when looking up the quoted reference [12] one is referred to yet other “detailed descriptions” in which again “detailed descriptions” in other preceding publications are quoted. I gave up going through a history of publications of the group. I suggest that some essential points (even if they repeat previous discussions) should be at least briefly addressed and then specific references to conclusive previous work can be provided.

Here are some questions about the experiment that occurred to me when reading the manuscript.

1.       An electrostatic separator can only separate ions from neutrals and act as an energy filter. Ions with different charge states or masses (e.g. the thallium isotopes) cannot be separated from one another. I am sure that the ion source produces other ions along with Tl+. A few words at least are necessary to explain the situation and the possible influence on the measured signals.

2.        How is the energy spread of the electron beam controlled?

3.       There does not appear to be a control of the electron-  and ion-beam overlap which may change with changing electron energy.

4.       Similarly, the determination of the possibly energy-dependent solid angle of photon detection is a fundamental issue.

5.       Statistical and systematic uncertainties of the measured cross sections should be clearly specified.

 Line 180; Fig. 2: The lines (black and gray) showing the theoretical results can be hardly distinguished. The meaning of the three tick marks between 5 and 10 eV is not clear. What is the meaning of the error bars in the figure?

 Line 218: There are not only energy levels of Tl+ but also of neutral Tl.

 Line 228: Instead of “Starting from” I suggest “At electron energies above”

Line 268: These AIS converge to

 Lines 280 and following: Part of this should already be addressed in the discussion of the experimental details.

 Lines 323 and following: Watch the number of electrons in each configuration given. In particular, the numbers of 5d electrons appear to be inconsistent with the possible ionic charge states addressed.

Author Response

Dear Referee,

Thank you for your time and valuable comments on our manuscript. Please see below our response to your comments.

Comment. Line 14: “to” should probably be replaced by “from subsequent(?)”. The sentence is not well phrased and thus, somewhat misleading. When ionic autoionizing states (of Tl+?) undergo electron decay (Auger decay?)  the final charge state is no longer Tl+ and therefore, no contributions from transitions in Tl+ can be observed in that case.

Response. Thank you for bringing this to our attention. Of course you are right, ionic autoionizing states can contribute to the cross-sections under study only due to radiative decay. In view of this, we rearrange this sentence in the revised version:

“A distinct structure revealed in the cross-sections of both lines results from electron decay of atomic autoionizing states and radiative transitions from upper ionic levels.”

Comment. Line 16: somewhere in the text (not necessarily in the Abstract), the term “effective cross section” should be defined.

Response. By the term “effective cross-section” we mean the “emission cross-section”. Taking into account that the term “emission cross-section” is a much more commonly used term and does not require further definition, we have replaced the “effective cross-section” by the “emission cross-section” throughout the whole text in the revised version.

Comment. Equation 1: misprint …hhh…

Response. We don't know but this is a result of the converting into a pdf format. In the source docx-file there is only one h.

Comment. Line 71: rather than describing all aspects of the experiment in detail, the manuscript refers to previous work. This is o.k., in principle. However, when looking up the quoted reference [12] one is referred to yet other “detailed descriptions” in which again “detailed descriptions” in other preceding publications are quoted. I gave up going through a history of publications of the group. I suggest that some essential points (even if they repeat previous discussions) should be at least briefly addressed and then specific references to conclusive previous work can be provided.

Response. We have made amendments to the experimental section according to your questions. See our responses below.

Comment. 1. An electrostatic separator can only separate ions from neutrals and act as an energy filter. Ions with different charge states or masses (e.g. the thallium isotopes) cannot be separated from one another. I am sure that the ion source produces other ions along with Tl+. A few words at least are necessary to explain the situation and the possible influence on the measured signals.

Response. We used a metallic thallium sample with 99.9% purity and with natural abundance of the isotopes 205Tl (70.48 %) and 203Tl (29.52 %) (from Sigma Aldrich). Therefore, impurity ions, taking into account their tiny percentage, do not influence the measured signals. Of course, the 90o electrostatic separator used in our experiment cannot separate the thallium isotopes. Therefore, the  measured signals are related to both isotopes. As for ions with different charge states, the use of the low-voltage arc discharge ion source (with a discharge voltage <10 V) allows us to obtain the beam of only single-charged Tl+ ions, moreover mainly in the ground state. To obtain Tl+2 ions, the discharge voltage not less than 26.5 V is needed.

In the revised version we added to the experimental section the sentence “A metallic thallium sample with 99.9% purity is used”.

Comment. 2. How is the energy spread of the electron beam controlled?

Response. In the revised version we have added the following text to the experimental section:

“The electron monoenergeticity depends on the accelerating voltage at which the electron gun is adjusted. It is 7 V (FWHM»0.5 eV) and 10 V (FWHM»1 eV). The adjustment is to set appropriate voltages on the first and second anodes of the electron gun so that to obtain the maximum electron current simultaneously with a current–voltage characteristic rapidly increasing and saturating as soon as possible. A retarding potential technique is used to determine the electron monoenergeticity.”

Comment. 3. There does not appear to be a control of the electron- and ion-beam overlap which may change with changing electron energy.

Response. In our experiment the electron beam (with a cross-section area of 1´8 mm2 mm2) is fully positioned within the ion beam cross-section (2.5´2.5 mm2). The stability of the beams overlap is provided by the stable alignment of the electron beam within the ion beam in the collision region with changing electron energy. This is monitored by the ratio of electron current to coaxially positioned external and internal Faraday cups made in the form of parallelepipeds with the entrance apertures of 3´12 mm2 and 2´10 mm2, respectively. We believe that the electron- and ion-beam overlap is not changed with changing electron energy.

In the revised version we have added the following text to the experimental section:

“The electron beam is fully positioned within the ion beam cross-section. The electrons that passed through the collision region are detected by coaxially positioned external and internal Faraday cups made in the form of parallelepipeds with the entrance apertures of 3´12 mm2 and 2´10 mm2, respectively. This allows us to monitor the stable alignment of the electron beam within the ion beam with the electron energy change by the constancy of the ratio of the currents to each of the cups.”

Comment. 4. Similarly, the determination of the possibly energy-dependent solid angle of photon detection is a fundamental issue.

Response. In the light of the above, we believe that the solid angle of photon detection is not practically changed with the electron energy change.

Comment. 5. Statistical and systematic uncertainties of the measured cross sections should be clearly specified.

Response. We have rearranged the last paragraph in the revised version:

“The electron beam energy and monoenergeticity are determined with uncertainty of not more than 0.1 eV. The electron energy scale is calibrated according to the excitation threshold of the λ121.6 nm line of atomic hydrogen by electron impact. The calibration uncertainty of the spectral sensitivity of the detection system, determined on the basis of the emission intensities of the atomic nitrogen spectral lines [13] resulting from the electron impact of N2 molecules measured at the electron energy of 100 eV, is about 16%.

The experimental uncertainties are dominated by the statistical uncertainties of counting. Therefore, the uncertainty of the useful signal measurement is evaluated with a standard uncertainty using the method of evaluation of uncertainty by the statistical analysis of a series of observations. All statistical uncertainties are quoted at the 68% confidence level (CL) corresponding to the mean standard deviation. The uncertainty of the relative measurement does not exceed 15% and 25% for the l190.8 nm and l179.9 nm lines, respectively. The total standard uncertainty of the absolute emission cross-section determination, involving the uncertainties of the relative measurement and the spectral sensitivity calibration, does not exceed 31% (l190.8 nm) and 41% (l179.9 nm).”

Comment. Line 180; Fig. 2: The lines (black and gray) showing the theoretical results can be hardly distinguished. The meaning of the three tick marks between 5 and 10 eV is not clear. What is the meaning of the error bars in the figure?

Response. For better visibility, the theoretical results are presented with solid (190.8 nm (RDW)) and dashed (179.9 nm (RDW)) curves in the revised version. Tick mark label “5” is now on his place in the revised version. We have added to the figure 2 caption that “Error bars are the uncertainty of the relative measurements”.

Comment. Line 218: There are not only energy levels of Tl+ but also of neutral Tl.

Response. We have added this to the figure 3 caption and the sentence referring to this figure.

Comment. Line 228: Instead of “Starting from” I suggest “At electron energies above”

Response. Thank you. We have changed this in the revised version.

Comment. Line 268: These AIS converge to

Response. Thank you. We have changed this in the revised version.

Comment. Lines 280 and following: Part of this should already be addressed in the discussion of the experimental details.

Response. This section is devoted to determination of the cross-section absolute values which is not the experimental procedure. The only moment concerning the experiment in this section is that “for correct comparison, the intensities of all three lines are measured at 100 eV in the same experiment”. We think it is more logically to say about this here rather than in the experimental section.

Comment. Lines 323 and following: Watch the number of electrons in each configuration given. In particular, the numbers of 5d electrons appear to be inconsistent with the possible ionic charge states addressed.

Response. Thank you. We have checked the number of electrons in each configuration given.

Reviewer 2 Report

-          The present manuscript presents important results for theoretical and experimental atomic structure data of Tl+ ion. I would accept it after minor revision,

-          A table containing energy levels should be included to the manuscript.

-          The transition 6s6p 3P1 – 6s2 1S0 (190.8) has been identified and presented in NIST, could the authors provide the superior of their results missed in previous works.

-          I think the novel part in this study is the experimentally measurements of the line 6s7s 1S0 – 6s6p 3P1, which should be focused on the introduction section.

-          In the second paragraph page 3, authors indicated to the uncertainty of measurements without referring to the method of uncertainty determination.

-          The RDW method has been used to calculate the effective cross section 3-transitions of 6?6? 1?1→6?2 1?0, 6?7? 1?0 →6?6? 1?1, and 6?2 1?2→6?6? 1?1. Figure 2, contains the effective cross section of 6s7s 1S0 – 6s6p 3P1 and 6s6p 3P1 – 6s2 1S0, where are the 6?7? 1?0 →6?6? 1?1, and 6?2 1?2→6?6? 1?1 transitions as a comparison to present results?

-          I can see some resonances in Fig 2, which means electron capture, would you shed light in such physical phenomenon. As well as the deviation between experiment and theory should be highlighted.

-          More details missed about the calculation’s procedures, in the way which how the authors adjusted the mesh points of partial and total cross sections calculations.

-          What about the correlation’s effects of the upper levels? How they affect the accuracy of the present calculations directly in energy levels and indirectly at the cross sections?

-          The relativistic effect are incorporated to the RDW method in the present calculations, more details are preferred in how this incorporations improved the present results and a comparison between the calculations with- and without adding the relativistic effects.

Author Response

Dear Referee,

Thank you for your time and valuable comments on our manuscript. Please see below our response to your comments.

Comment. A table containing energy levels should be included to the manuscript.

Response. We have added the table containing energy levels in the revised version.

Comment. The transition 6s6p 3P1 – 6s2 1S0 (190.8) has been identified and presented in NIST, could the authors provide the superior of their results missed in previous works.

Response. The NIST database provides only transition wavelength and radiative probability for the intercombination 6s6p 3P1 – 6s2 1S0 (190.8 nm) transition obtained from the lifetime of the 6s6p 3P1 level. In the present work we studied the electron-impact emission cross-section of the process 6s2 1S0 (Tl) +e -> 6s6p 3P1 (Tl+*)+e’ -> 6s2 1S0 (Tl) + 190.8 nm (see the process (1) the introduction section). So far, such data are not reported in the NIST.

Comment. I think the novel part in this study is the experimentally measurements of the line 6s7s 1S0 – 6s6p 3P1, which should be focused on the introduction section.

Response. In the revised version we have added the following sentenceat the end of the introduction section:

“Note that no data on the process (2) are available in the literature so far.”

Comment. In the second paragraph page 3, authors indicated to the uncertainty of measurements without referring to the method of uncertainty determination.

Response. We have rearranged the last paragraph concerning the uncertainty of measurements in the revised version:

“The electron beam energy and monoenergeticity are determined with uncertainty of not more than 0.1 eV. The electron energy scale is calibrated according to the excitation threshold of the λ121.6 nm line of atomic hydrogen by electron impact. The calibration uncertainty of the spectral sensitivity of the detection system, determined on the basis of the emission intensities of the atomic nitrogen spectral lines [13] resulting from the electron impact of N2 molecules measured at the electron energy of 100 eV, is about 16%.

The experimental uncertainties are dominated by the statistical uncertainties of counting. Therefore, the uncertainty of the useful signal measurement is evaluated with a standard uncertainty using the method of evaluation of uncertainty by the statistical analysis of a series of observations. All statistical uncertainties are quoted at the 68% confidence level (CL) corresponding to the mean standard deviation. The uncertainty of the relative measurement does not exceed 15% and 25% for the l190.8 nm and l179.9 nm lines, respectively. The total standard uncertainty of the absolute emission cross-section determination, involving the uncertainties of the relative measurement and the spectral sensitivity calibration, does not exceed 31% (l190.8 nm) and 41% (l179.9 nm).”

Comment. The RDW method has been used to calculate the effective cross section 3-transitions of 6?6? 1?1→6?2 1?0, 6?7? 1?0 →6?6? 1?1, and 6?2 1?2→6?6? 1?1. Figure 2, contains the effective cross section of 6s7s 1S0 – 6s6p 3P1 and 6s6p 3P1 – 6s2 1S0, where are the 6?7? 1?0 →6?6? 1?1, and 6?2 1?2→6?6? 1?1 transitions as a comparison to present results?

Response. The present work is focused only on the study of the intercombination 6s6p 3P1 – 6s2 1S0 and 6s7s 1S0 – 6s6p 3P1 transitions. Therefore, comparison of the RDW cross-sections for these two transitions with the present measurements is shown in Figure 2. As for the 6?6? 1?1→6?2 1?0, 6?7? 1?0 →6?6? 1?1, and 6?2 1?2→6?6? 1?1 transitions, they are studied in our previous work [11]. In the present work we only used the absolute value of the cross-section for the 6?6? 1?1→6?2 1?0 transition to normalize our measurement.

Comment. I can see some resonances in Fig 2, which means electron capture, would you shed light in such physical phenomenon. As well as the deviation between experiment and theory should be highlighted.

Response. You are right. Some resonances in Fig 2 are due to the electron capture. We have added the following text to the section 4.1 (paragraph 3) for clarity:

“In the near-threshold energy range, below the energy of the first excited 6s7s 3S1 level (»13 eV) from which the transition to the  level is already possible, the structure is related to the resonance excitation involving resonance capture of an incident electron with simultaneous excitation of one of the ion electrons (dielectronic capture) resulting in the formation of an atomic AIS which subsequently decay to an excited state of the ion in the electron channel:

e + Tl+® Tl** ® Tl+* + e,

where Tl** is atomic AIS, Tl+* is an excited state of the ion.”

As for the deviation between experiment and theory, the reason for this is discussed in the manuscript (lines 299-302) as follows:

“However, the structure is missing in the theoretical cross-section since other inelastic channels, such as autoionization, electron attachment, etc., are not included in the RDW calculations. Furthermore, the calculations are performed in the first order RDW approximation.”

Comment. More details missed about the calculation’s procedures, in the way which how the authors adjusted the mesh points of partial and total cross sections calculations.

Response. We have added the text below in Section 3 Theoretical Method:

“In the atomic wavefunction calculations the GRASP2018 gives the radial wave functions up to 1990 points using an exponential grid. The radial wavefunctions are further interpolated at 7000 mesh points using the RDW program and thereafter, distorted wavefunctions for projectile and scattered electrons are also calculated at the same 7000 grid points. The RDW program can handle up to 250 partial waves and for a given incident electron energy it uses as many partial waves as are required to achieve the tolerance of 10-5 in the evaluation of the T-matrix. Once the convergence is reached, the excitation cross section is finally determined from Equation (3). More details of the RDW method can be found in [19]. “

Comment. What about the correlation’s effects of the upper levels? How they affect the accuracy of the present calculations directly in energy levels and indirectly at the cross sections?

Response. The correlation effects play an important role and must be included to determine MCDF wavefunctions of heavy ions like Tl+. In the present MCDF calculations, we have included core-valence and valence- valence correlations. Our procedure to obtain the MCDF wavefunctions is mentioned in the submitted manuscript on Line 124-132. We carried out MCDF calculations using different choices of MR sets and active spaces. Based on the comparison of the energy levels with the corresponding values from the NIST database, the best optimized wavefunctions are used in the RDW calculations to obtain the distorted wavefunctions. The accuracy of the MCDF wavefunctions is discussed on Page 4, line 133 – 146.

These two types of (MCDF and distorted) wavefunctions are the main ingredient in the cross-section calculations. Hence, the accuracy of the MCDF wavefunctions has direct impact on the accuracy of the cross-sections.

Comment. The relativistic effect are incorporated to the RDW method in the present calculations, more details are preferred in how this incorporations improved the present results and a comparison between the calculations with- and without adding the relativistic effects.

Response. The relativistic effects are incorporated directly in the RDW method by solving the Dirac equations for both, the bound and continuum electrons. Thus, the Dirac equation takes care of the relativistic effects by default and the corrections like spin-orbit interaction, Darwin terms etc. are not required to be used as perturbation.

We are afraid that the comparison between non-relativistic and relativistic calculations is not possible here. Carrying out non-relativistic distorted wave calculations is not trivial and is itself an elaborated task. Since Tl+ contains 80 electrons and its nuclear charge is 81, a relativistic treatment through Dirac equation is essential for such a heavy element. Hence, a relativistic theory is used in the present study.